# Selective Flamingo Medium for the Isolation of *Aspergillus fumigatus*

**DOI:** 10.3390/microorganisms9061155

**Published:** 2021-05-27

**Authors:** Jianhua Zhang, Alfons J. M. Debets, Paul E. Verweij, Sijmen E. Schoustra

**Affiliations:** 1Laboratory for Genetics, Wageningen University and Research, 6708 PB Wageningen, The Netherlands; fons.debets@wur.nl (A.J.M.D.); sijmen.schoustra@wur.nl (S.E.S.); 2Department of Medical Microbiology, Radboud University Medical Center, 6500 HB Nijmegen, The Netherlands; paul.verweij@radboudumc.nl; 3Center of Expertise in Mycology Radboudumc/CWZ, 6500 HB Nijmegen, The Netherlands; 4Center for Infectious Diseases Research, Diagnostics and Laboratory Surveillance, National Institute for Public Health and the Environment (RIVM), 3720 BA Bilthoven, The Netherlands

**Keywords:** *Aspergillus fumigatus*, environmental samples, selective medium, Mucorales

## Abstract

For various studies in the clinic as well as the environment, it is essential to be able to selectively isolate *Aspergillus fumigatus* from samples containing bacteria as well as various other fungi (mainly Mucorales). Six agar media were compared for effectiveness in selectively isolating *Aspergillus fumigatus* from agricultural plant waste, woodchip waste, green waste, soil, grass and air samples collected in The Netherlands at a 48 °C incubation. The Flamingo Medium incubated at 48 °C, provided the most effective condition for the isolation of *A. fumigatus* from environmental samples, since it effectively inhibited the growth of competing fungi (mainly Mucorales) present in the environmental samples. Flamingo Medium reduced the number of colonies of Mucorales species by 95% and recovered an average of 20−30% more *A. fumigatus* colonies compared to the other media. We further confirmed that Flamingo Medium can inhibit the growth of clinical Mucorales, which occasionally present in patient’s tissue and can also be used for clinical applications. We suggest the use of Flamingo Medium as an efficient method for the study of *A. fumigatus* from important environmental niches for which there is increasing interest. Additionally, it can also be used in the clinic to isolate *A. fumigatus* especially from tissue contaminated with Mucorales.

## 1. Introduction

*Aspergillus fumigatus* is a common plant-waste-degrading fungus of which spores are abundantly present in the air. When inhaled, these spores can cause diseases in animals and humans ranging from allergic syndromes to acute invasive aspergillosis depending on the host immune system [1,2]. The number of drug classes that are available for the treatment of *Aspergillus* diseases remains very limited, with the azoles representing the major class. Azole resistance is an emerging concern in *A. fumigatus*, complicating the treatment of patients [3,4]. Up to 90% of *A. fumigatus* isolates recovered from patients with azole-resistant invasive aspergillosis exhibit resistance mutations that are associated with resistance selection in the environment [2,5,6,7,8]. The health importance of *A. fumigatus* and safety of azole fungicide application in agriculture have led to the increased interest in the study of *A. fumigatus* from environmental samples [9,10]. In previous research we have gathered evidence that azole-resistant *A. fumigatus* can accumulate and thrive in plant waste that contains agricultural azole residues [6,11]. The concept of a hotspot for azole-resistant *A. fumigatus* was postulated, which is characterized by an environment that supports the growth and reproduction of *A. fumigatus* and where azole fungicides with anti-*Aspergillus* activity are present [6]. In an initial survey, three hotspots were identified in The Netherlands: decaying flower bulb waste from farms, industrial wood-chip waste and industrial green-waste storage [6]. It is likely that additional hotspots will be identified over time, where both conditions are present.

For the identification of hotspots an efficient method for unbiased and quantitative isolation of *A. fumigatus* from environmental samples is crucial. This isolation of *A. fumigatus* has commonly been performed in several studies by using selective and non-selective media, including Malt Extract Agar (MEA) supplemented with Chloramphenicol and Streptomycin (MEA+C+S), Sabouraud Detrose Agar supplemented with Chloramphenicol (SDA+C) and Dichloran-Glycerol (DG18) [11,12,13,14]. On these and other media growth of Mucorales species (~10^3−6^ Colony Forming Units/mL (CFU/mL) compromises the usefulness of MEA+C+S and DG18 [14,15,16]. Culturing at 48 °C is a generally applied condition to isolate *A. fumigatus*, since it restricts the growth of many fungi. However, environmental samples may contain a large number of thermophilic Mucorales species isolates hampering the isolation of *A. fumigatus* [17,18]. A good example of a mucoraceous thermophile is *Rhizomucor pusillus* with a maximum growth temperature of 54–58 °C. Another example of thermophilic Mucorales is *Lichtheimia corymbifera*, which can grow up to 45–50 °C [19]. We previously found that approximately 50% of the plant-waste samples from which *A. fumigatus* was isolated, also contained Mucorales [6,20]. Thus, an efficient selective growth medium to facilitate selective isolation of *A. fumigatus* from environmental samples is lacking.

In this present study, we aimed to develop a selective medium for the isolation of *A. fumigatus* from environmental samples. We have investigated and compared six types of media for their properties of selectively isolating *A. fumigatus* from samples that also contain high levels of spores or mycelia from Mucorales species. Of these six, three media (MEA+C+S, SDA+C and DG18) are commonly used for isolating *A. fumigatus.* We further included Modified Rose Bengal Agar (M-RB), which has originally been developed for the isolation of *A. flavus* and two modified media that have not been evaluated before (Flamingo Medium and MEA-Rose Bengal). In addition, we tested the best of these 6 media for its properties for selective isolation of *A. fumigatus* from clinical samples that contain both *A. fumigatus* and Mucorales species.

## 2. Materials and Methods

### 2.1. Media Preparation

MEA+C+S: Suspend 30 g of Malt Extra Agar (MEA) (Sigma-Aldrich, Steinheim, Germany) and 15 g of agar in 1 L of distilled water, autoclave at 121 °C for 15 min. Supplement with 1 mL of 50 mg/mL of Chloramphenicol and Streptomycin (Sigma-Aldrich, Steinheim, Germany) before use.

SDA+C: Suspend 10 g of Sabouraud Glucose Agar (Sigma-Aldrich, Steinheim, Germany) in 1 L of distilled water, autoclave at 121 °C for 15 min. Supplement with 1 mL of 50 mg/mL of Chloramphenicol (Sigma-Aldrich, Steinheim, Germany) before use.

DG18 (Sigma-Aldrich, Steinheim, Germany): Suspend 31.6 g of medium and 220 g of glycerol in 1 L of distilled water, autoclave at 121 °C for 15 min.

M-RB: Suspend 3 g of sucrose and NaNO_3_; 0.3 g of KH_2_PO_4_; 0.5 g of MgSO_4_·7H_2_O and KCl; 0.7 g of K_2_HPO_4_; 10 g of NaCl and 1 mL of Adye & Mateles Reagent stock X1000/mL dH_2_O(0.7 mg Na_2_B_4_O_7_·10H_2_O; 0.5 mg of (NH_4_)_6_MO_7_O_24_·4H_2_O; 10 mg of Fe_2_(SO_4_)_3_·6H_2_O; 0.3 mg CuSO_4_·5H_2_O; 0.11 mg of MnSO_4_·H_2_O; 17.5 mg of ZnSO_4_·7H_2_O) into 1 L of distilled water; autoclave at 121 °C for 15 min. Supplement with 1 mL of 50 mg/mL of Chloramphenicol and Streptomycin (Sigma-Aldrich, Steinheim, Germany), and 5 mL 5 mg/mL Rose Bengal and 10 mL of 1 mg/mL of Dichloran before use [21].

Flamingo Medium: Suspend 6.0 g NaNO_3_, 1.5 g KH_2_PO_4_, 0.5 g MgSO_4_. 7H_2_O, 0.5 g KCl, 10 mg of FeSO_4_, ZnSO_4_, MnCl_2_ and CuSO_4_ and agar 15 g in 1 L of distilled water (adjust pH to 5.8), autoclave at 121 °C for 15 min. Supplement with 5 mL 5 mg/mL Rose Bengal 10 mL of 1 mg/mL of Dichloran and 1 mL of 50 mg/mL of Chloramphenicol and Streptomycin (Sigma-Aldrich, Steinheim, Germany) before use (see Appendix B).

MEA-RB: Suspend 30 g Malt Extra Agar (Sigma-Aldrich, Steinheim, Germany) and 15 g of agar in 1 L of distilled water, autoclave at 121 °C for 15 min. Supplement with 5 mL 5 mg/mL Rose Bengal, 10 mL of 1 mg/mL of Dichloran, 1 mL of 50 mg/mL of Chloramphenicol and Streptomycin (Sigma-Aldrich, Steinheim, Germany) before use.

Collection of environmental samples: Soil samples, wood chips and green-waste samples were obtained from a previous study [22]. Plant-waste samples (decaying flower bulb waste) were collected from a farm in the province of North-Holland, The Netherlands. Samples were taken at least 100 m apart.

Air samples were collected in Wageningen using a Coriolis air sampler (Bertin, France) with a setting of 200 L/min during 2 min. Samples were taken at least 100 m apart. The airborne particles were dissolved into 15 mL of saline with 0.05% Tween 80 in a plastic cone. After collection, the liquid was used for plating on the various growth media. For the samples without visible colonies, the liquid was concentrated with 5000 rpm centrifuging (Germany) for 1 min, 10 mL supernatant was removed, and the sediment was suspended in 5 mL liquid, which was used for replating.

### 2.2. Isolating A. fumigatus Using Various Media

For each sample of plant waste, wood chips, green waste, and soil, 5 g was added to 10 mL sterile saline (0.8 g/L NaCl in water) with 0.05% Tween 80 and diluted. As stated above, air samples were collected in 5 mL saline. After vortexing for 2 min, 50 µL of a diluted suspension (10^0^ to10^−3^) was plated on the six test media (MEA+C+S, SDA+C, DG18 and Flamingo, M-RB, and MEA-RB). Three replicates were applied for each medium. Cultures were incubated at 48 °C, which is generally used for selective growth of *A. fumigatus* [20] After three days of incubation, colonies of *A. fumigatus* and surface area covered by Mucorales species on MEA+C+S, SDA+C; MEA-RB and Flamingo medium were recorded. DG18 and M-RB plates were recorded after five days because *A. fumigatus* colonies were not easily recognized after three days on DG18 medium, and *A. fumigatus* colonies were too small and did not yet sporulate on M-RB medium. The colonies that showed *Aspergillus* morphology were selected and verified for *A. fumigatus* molecular characteristics by amplifying (PCR) and sequencing part of the ß-tubulin and carboxypeptidase-5 genes [8,12]. The genes encoding β-tubulin and carboxypeptidase-5 were amplified with the primer sets benA (forward, 5′-AATTGGTGCCGCTTTCTGG-3′; reverse, 5′-AGTTGTCGGGACGGAATAG-3′) and cxp (forward, 5′-GAACATTAGCCCCAGTTGAG-3′; reverse primer, 5′-CACTTCTTCTTGCACGTAGTC-3′), respectively. The amplified DNA fragments were purified with ExoSAP-IT™ PCR Product Cleanup Reagent (Thermo Fisher (Waltham, MA, USA)). DNA sequencing of the forward strand of each fragment was performed at the Eurofins Genomics (Ebersberg, Germany). The resulting sequences were aligned in CLUSTALW46 using the program BioEdit47.

### 2.3. Validation of Flamingo Medium

We tested whether the Flamingo Medium allows for quantitative isolation of *A. fumigatus* unbiased for known specific genotypes by plating different genotypes (three replicates) of *A. fumigatus*, either in mixed population or as single culture on both MEA and Flamingo Medium, after three days of growth at 48 °C, the total number of colonies were counted. Additionally, the selectivity of Flamingo Medium for *A. fumigatus* was further confirmed by testing artificial mixtures of clinical *A. fumigatus* and clinical Mucorales. The clinical Mucorales fungi included *Rhizomucor pusillus* (V103-44); *Rhizopus arhieus* (V204-34); *Rhizopus microsporus* (V154-27); *Lichtheimia corymbifera* (V250-74) and were cultured from patients and stored in the fungus culture collection at the Radboud University Medical Centre. If patient samples contain both Mucorales species and *A. fumigatus*, the latter is commonly overgrown by the Mucorales fungus, which may preclude species identification and in vitro susceptibility testing. As both fungi are thermophilic, separation by incubation at high temperature is not possible. Of each Mucorales species 50 µL of a spore suspension (concentration of 1000 CFU/g) was mixed with 50 µL of a spore suspension of clinical isolate *A. fumigatus* V30-40 (concentration of 1000 CFU/g) and plated on the DG18 and Flamingo Medium. After three-day incubation at 48 °C, the number of *A. fumigatus* was recorded. The areas covered by fungi from the Mucorales group was measured using a transparent plastic format as shown in Appendix C. Each area has defined surface size. By placing this transparent plastic format under bottom, the plates with fungal colonies, the area covered by non- *A. fumigatus* (fungi of Mucorales group) was estimated by adding up all grids covered by these colonies.

Statistical analyses: Data distribution analyses were performed via SPSS-Analyze-Descriptive Statistics-Explore Plots-Histogram. Significance tests for differences in detected colonies among media in detection of *A. fumigatus* and surface area covered by non-*A. fumigatus* (Mucorales) were performed with a Kruskal–Wallis test. Differences between the total *A. fumigatus* CFUs on the Flamingo Medium and MEA were tested for statistically significant differences with a pairwise t-test.

## 3. Results

### 3.1. Flamingo Medium Is Highly Effective for the Isolation of A. fumigatus

Figure 1 shows counts of *A. fumigatus* (above) and surface area covered by non-*A. fumigatus* maily Mucorales (below, Appendix D) of various samples plated on the six media that we compared for selectivity of *A. fumigatus* isolation. Of six media compared, on average Flamingo Medium was the most efficient isolation medium. It reduced the number of Mucorales colonies by 95% and isolated 20−30% more *A. fumigatus* colonies compared to the other media in a short incubation time of three days (Figure 1, Appendix A with detailed information). MEA+C+S, SDA+C; MEA-RB and Flamingo Medium allowed for fast germination and growth of *A. fumigatus* after three days of incubation, Flamingo Medium best limited growth of most Mucorales species. M-RB has the same limitation of Mucorales species growth as Flamingo Medium, however, it required 5–7 days for *A. fumigatus* colonies to fully grow. Moreover, all colonies on the Flamingo Medium, but not all colonies on M-RB, were able to sufficiently sporulate for further culturing and testing. Carboxypeptidase-5 genes sequencing of colonies with different morphology on Flamingo Medium confirmed that all isolates are *A. fumigatus.* Figure 2 shows the isolation plates of *A. fumigatus* from one of plant-waste samples (ID: S127) on six different media.

### 3.2. Flamingo Medium Allows Rapid Growth for a Range of Genotypes of A. fumigatus

A series of common previously isolated environmental *A. fumigatus* genotypes (see Appendix E) were plated on MEA and Flamingo Medium in three replicates. We tested whether total CFUs on the Flamingo Medium were significantly different from the total counts on MEA by using a pairwise t-test after three days of incubation at 48 °C. The total number of *A. fumigatus* detected did not differ between the Flamingo and MEA media, both when plating a natural mixed population, and different single genotypes with different Cytochrome P450 14-alpha sterol demethylase (cyp51A) mutations.

### 3.3. RB and Dichloran Are Key Components in Flamingo Medium

Flamingo Medium can efficiently and selectively isolate *A. fumigatus* from environmental samples and this medium effectively inhibited growth of 95% Mucorales from agricultural samples when these were plated on the Flamingo Medium, illustrating that the Falmingo Medium is selective. To investigate which element is critical for inhibiting the growth of Mucorales in Flamingo Medium, RB and Dichloran were removed step by step. As shown in Figure 3B,C, Mucorales were able to grow on the medium when either RB or Dichloran was omitted, suggesting that both RB and Dichloran are essential components of the Flamingo Medium to achieve the inhibitory effect.

While on Flamingo Medium the growth of Mucorales fungi is greatly reduced, on 50% of the environmental samples (mainly soil, grass, wood chips) growth of Mucorales is still faintly visible, such as grass/root sample 143 (blue arrow Figure 3D). After introducing the antibiotics of Chloramphenicol and Streptomycin into Flamingo Medium, for 22% out of 50% of environmental samples, growth of Mucorales was completely inhibited when plated. Therefore, we conclude that the combination of RB, Dichloran, Chloramphenicol and Streptomycin effectively inhibits growth of Mucorales species from the environmental samples and makes the Flamingo Medium completely selective.

### 3.4. The Validation of Flamingo Medium for Isolating Clinical A. fumigatus from an Artificial Mixture of Clinical Isolates of A. fumigatus and Mucorales

We prepared mixtures of Mucolares fungi with and without an *A. fumigatus* strain. These mixtures were plated on both DG18 and on Flamingo Medium. After incubation for three days at 48 °C, results showed that Mucorales fungi grew on DG18 and not on Flamingo Medium, while *A. fumigatus* grew on both media. Results are illustrated in Figure 4.

## 4. Discussion

This study compared six growth media for their effectiveness in isolating *A. fumigatus* from environmental samples in which other thermophilic fungi were present, including Mucorales species. We developed a new medium, called Flamingo Medium. Flamingo Medium produced the best results for the quantitative isolation of *A. fumigatus* isolates from environmental samples irrespective of their genotype, while suppressing the growth of Mucorales species tested. Growth reduction of Mucorales species was on average 95% lower on Flamingo Medium than on MEA, SDA and DG18.

Some special features of the Flamingo Medium may explain its effectiveness. (1) Compared with commonly used growth media such as MEA, SDA, and DG18, Flamingo Medium provides nitrate as sole nitrogen source, which directly limits Mucorales species growth [21]. (2) Another essential component in Flamingo Medium is Rose Bengal. This compound strongly inhibits the fast-growing Mucorales. Further, Rose Bengal is a selective agent that inhibits bacterial growth and restricts the size and height of colonies of the more rapidly growing molds [23]. We demonstrated here that Rose Bengal allows for the growth of *A. fumigatus* irrespective of its genotype (i.e., of the 11 genotypes tested). (3) Another crucial compound is Dichloran, as this compound affects the colony diameter and enumeration of fungi, leading to tiny colonies on the plates. This can largely suppress the fast-growing fungi such as Mucorales [24].

Antibiotics Chloramphenicol and Streptomycin are generally supplemented in the medium to suppress bacterial growth from environmental samples when isolating fungi [20,21]. These two antibiotics play a critical role especially when isolating fungi at the temperature ranging from 25 to 37 °C. When isolating *A. fumigatus* at the temperature of 48 °C with Flamingo Medium, Chloramphenicol and Streptomycin were not considered critical elements, which was also confirmed by all environmental samples. Bacterial growth was not observed on Flamingo Medium without Chloramphenicol and Streptomycin. However, Mucorales colonies appeared on the plates in 50% of the samples. By adding these two antibiotics, 10% of the samples became Mucorales-free. Therefore, these two antibiotics in combination with Rose Bengal and Dichloran not only inhibit growth of bacteria, but also that of Mucorales. Furthermore, since the Mucorales group consists of a large variation in species, not all Mucorales species were completely inhibited on the Flamingo Medium (such as Figure 3D), but all were suppressed to small colonies, which does not influence the isolation of *A. fumigatus* in general.

Separating different fungal species is critical to allow study of individual species in mixed habitats. Composting vegetation represents a habitat selective for thermophilic fungi including *A. fumigatus* and some Mucorales. Both are considered to play an important role in the decomposition of cellulose, and other more recalcitrant plant material [25]. Although incubation at 48 °C is effective for culturing *A. fumigatus*, the presence of Mucorales species in environmental samples precludes selection of individual *A. fumigatus* colonies. As Mucorales are fast-growing thermotolerant fungi *A. fumigatus* is rapidly overgrown. Flamingo medium provides an effective way to suppress the growth of various Mucorales species, while supporting the growth of *A. fumigatus*. Flamingo Medium can therefore be widely used for isolating *A. fumigatus* from all sorts of environmental samples. Directly obtaining pure *A. fumigatus* isolates is simplified by omitting the extra steps of purification. Additionally, we found that the Flamingo Medium can be used for isolating other fungi at lower temperatures, such as *A. niger* and *A. flavus* at 30 °C and 37 °C, respectively (data not shown).

In addition to environmental studies, Flamingo Medium might also be useful in clinical mycology for occasional cases where *Aspergillus* and Mucorales co-infections are observed in patients with invasive fungal diseases. In addition to the identification of the fungal pathogens, in vitro susceptibility testing is important in *A. fumigatus*. In-host and environmental resistance selection has been widely published, especially against medical triazoles. As a pure culture is required for in vitro susceptibility testing Flamingo Medium may be useful to select *A. fumigatus*. Indeed, separating Mucorales from *A. fumigatus* was successful, when clinical isolates were used.

Developing this method for unbiased quantitative environmental sampling of *A. fumigatus* is of significant importance. It will allow for more effective monitoring of *A. fumigatus* in any sort of environmental niches, and further understanding of the origins and spread of *A. fumigatus*. This will facilitate isolation of environmental *A. fumigatus* in the context of hot spot research for the development of azole resistance and various other research where the isolation of *A. fumigatus* from the environment is crucial. Additionally, Flamingo medium exhibited the potential of selectively isolating *A. fumigatus* from clinical samples where Mucorales species are present that commonly compromise efficient *A. fumigatus* isolation. This may be of use when *A. fumigatus* from patient material that also harbors other fungi.

## Figures and Tables

**Figure 1 microorganisms-09-01155-f001:**
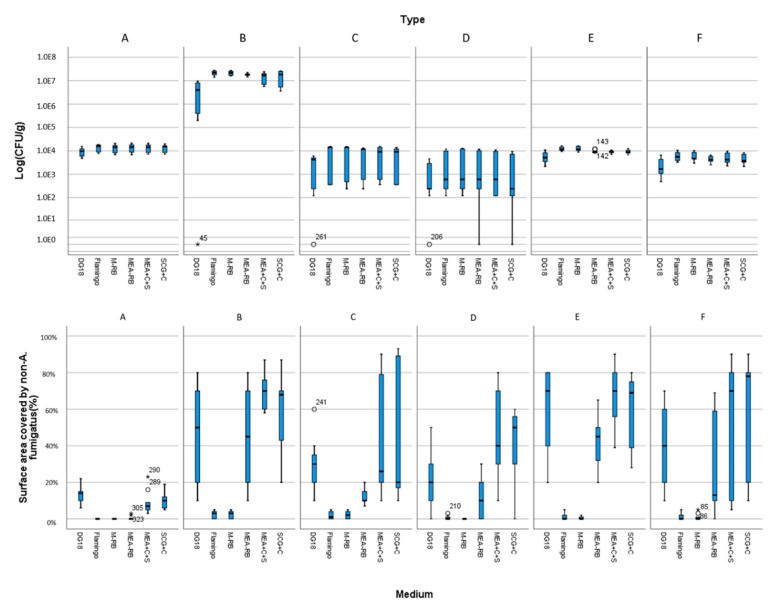
The total count of *A. fumigatus* detected in various samples (above) and the surface area of agar plates covered with Mucorales (below) using six different isolation media for each sample. Three biological and three technical replicates were used for each measurement. (**A**) Air; (**B**) plant waste; (**C**) ditch water; (**D**) grass/root; (**E**) soil; and (**F**) wood.

**Figure 2 microorganisms-09-01155-f002:**
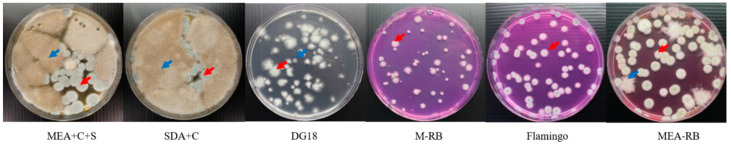
The cultures grown from plant-waste sample 127 on six growth media. I MEA+C+S, SDA+C, DG18, flamingo medium (after three days of incubation at 48 °C), DG18, and M-RB (after five days of incubation at 48 °C). The red arrow points to a typical *A. fumigatus* colony; the blue arrow points to a typical colony of a Mucorales species.

**Figure 3 microorganisms-09-01155-f003:**
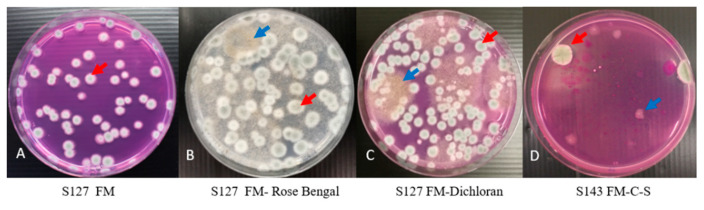
Culturing of *A. fumigatus* from plant-waste sample 127 on Flamingo Medium (**A**), and Flamingo Medium without Rose Bengal (**B**) and Flamingo Medium without Dichloran (**C**). The culture of *A. fumigatus* from grass/root sample 143 on the Flamingo Medium without Chloramphenicol and Streptomycin is shown in panel (**D**). All plates were incubated for three days at 48 °C. The red arrow points to a typical *A. fumigatus* colony; the blue arrow points to a typical colony of a Mucorales species. FM: Flamingo Medium.

**Figure 4 microorganisms-09-01155-f004:**
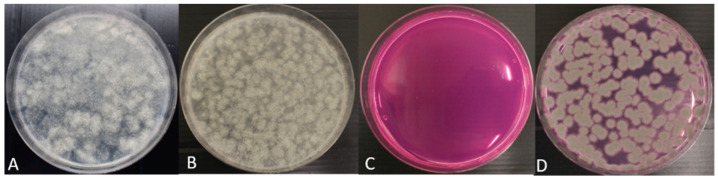
The validation of Flamingo Medium for isolating clinical *A. fumigatus* by artificially mixing clinical *A. fumigatus* and clinical Mucorales species. (**A**) mixed clinical Mucorales species on DG18; (**B**) mixed Mucorales species and *A. fumigatus* (V30-40) on DG18. (**C**) mixed Mucorales species on Flamingo Medium. (**D**) mixed Mucorales species and *A. fumigatus* (V30-40) on Flamingo Medium. Mixture of Mucorales species consisted of *Rhizomucor pusillus* (V103-44), *Rhizopus arhieus* (V204-34), *Rhizopus microsporus* (V154-27), and *Lichtheimia corymbifera* (V250-74).

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
