# Peer review of "Selective Flamingo Medium for the Isolation of Aspergillus fumigatus"

_microorganisms, 2021, doi:10.3390/microorganisms9061155_

Round 1

Reviewer 1 Report

The manuscript presents a culture medium to effectively and easily isolate A. fumigatus from other microorganisms in a variety of samples.

Throughout the text the authors refer to the ability to separate individual colonies of A. fumigatus from bacteria and other fungi, but the results shown are limited to Mucorales. Some species are mentioned in Fig 4 and other fungi are mentioned in the discussion, but it would give a much better picture of the usefulness and quality of the Flamingo medium to show the wider capacity of discrimination.

Following from the above, it would be very informative to provide an idea of the type and variety of organisms found in the environmental samples chosen and then narrow these down to the types of fungi of main interest.

One last comment is that though the clinical sample mixtures are revealing, it would be very informative to see how real unknown clinical samples behave.

Specific comments

METHODS

Clm report link missing?

The methods for the verification of Aspergillus by PCR amplification and sequencing are missing. These should be specifically described and presented as a separate section, providing primer sequences, kits and machines used.

How were the areas covered by the different organisms measured?

RESULTS

Abbreviations such as CFU and Cyp51 need to be specified the first time they appear in the text. (there are several other, like FM, MM, etc).

Just as DG18 and M-RB media were incubated longer (5 days) than the other to achieve growth, it would be informative to culture MEA+C+S and SDA+C for shorter periods of time than 3 days to allow distinguishing between the colonies.

In Fig 2, there seem to be colonies different from both Aspergillus and Mucorales, by size or colour, and it would be interesting to know whether these are indeed different organisms or strains or not. A more detailed description of this section would be helpful.

The last sentence of the first paragraph in section 3.3 doesn’t make sense, would these compounds be essential to stop Mucorales from growing? This needs to be explained more clearly. Similarly, the before last sentence of the next paragraph states that the addition of antibiotics makes 10% of the samples Mucorales-free. Does this refer to 10% of the previous 50% or in general, this needs to be explained better.

Section 4.3 is 1 sentence that is very long and hard to follow. It should be broken down into shorter statements and the results explained better.

I did not see any supplementary figures available on the system, only tables. A supplementary Fig 2 is mentioned in the first sentence of the results section.

Author Response

REVIEW 1

The manuscript presents a culture medium to effectively and easily isolate A. fumigatus from other microorganisms in a variety of samples.

Throughout the text the authors refer to the ability to separate individual colonies of A. fumigatus from bacteria and other fungi, but the results shown are limited to Mucorales. Some species are mentioned in Fig 4 and other fungi are mentioned in the discussion, but it would give a much better picture of the usefulness and quality of the Flamingo medium to show the wider capacity of discrimination.

Reply: Thank you for your comments. yes, you are right, in this manuscript, we mainly focus on removing the group of Mucorales, as we noted in the INTRO Line 61-64. We chose this approach since fungi of the Mucorales group are the main fungi found in environmental and clinical samples alongside of Aspergillus fumigatus  at 48 degree incubation. The growth of these Mucorales fungi impedes with the isolation of A. fumigatus from these samples and when developing a selective medium for A. fumigatus, it is key to restrict the growth of fungi from the Mucorales group. 

Additionally, in Appendix C, we have now provided the picture of fungi of the Mucorales group that grew during the isolation A. fumigatus on non-selective media from various environmental samples (shown as medium name- Sample ID).

you can find Appendix C in main ms, because here doesnot pictures format 

Following from the above, it would be very informative to provide an idea of the type and variety of organisms found in the environmental samples chosen and then narrow these down to the types of fungi of main interest.

Reply: Our previous work has shown that when incubated at 48 oC, only fungi from the Mucorales group as well as Aspergillus fumigatus will be able to grow (such as Rhizomucor pusillus with a maximum growth temperature of 54–58°C. Other examples of thermophilic Mucorales include Lichtheimia corymbifera which can grow up to 45–50°C in Line 63-65). This is why we focussed on these fungal groups when developing our selective media. As the Reviewer indicates, environmental samples will contain a very large range of fungi https://science.sciencemag.org/content/346/6213/1256688.abstract). Yet, because most of these are unable to grow at 48 oC, these were not included in our study.

One last comment is that though the clinical sample mixtures are revealing, it would be very informative to see how real unknown clinical samples behave.

Reply: Thank you for your comments. We indeed agree that testing the medium on unknown clinical samples would be optimal, but these samples are relatively uncommon. Despite having the medium available in a clinical microbiology laboratory for 6 months, such samples were not encountered. We therefore choose to demonstrate its potential on clinical cultures – which are representative for what we may encounter in (real/unknown) clinical samples.

Specific comments

METHODS

Clm report link missing?

Reply: It has been inserted in L106. For unknown reason, the latest version with link did not show up in the submission system. Sorry for this. It has been corrected.

The methods for the verification of Aspergillus by PCR amplification and sequencing are missing. These should be specifically described and presented as a separate section, providing primer sequences, kits and machines used.

Reply: The methods for the verification of Aspergillus by PCR amplification and sequencing are now shown in the text L127-136. “The colonies that showed Aspergillus morphology were selected and verified for A. fumigatus molecular characteristics by amplifying (PCR) and sequencing part of the ß-tubulin and carboxypeptidase-5 genes (8, 12).” The genes encoding β-tubulin and carboxypeptidase-5 were amplified with the primer sets benA (forward, 5′-AATTGGTGCCGCTTTCTGG-3′; reverse, 5′-AGTTGTCGGGACGGAATAG-3′) and cxp (forward, 5′-GAACATTAGCCCCAGTTGAG-3′; reverse primer, 5′-CACTTCTTCTTGCACGTAGTC-3′), respectively. The amplified DNA fragments were purified with a ExoSAP-IT™ PCR Product Cleanup Reagent (Thermo Fisher). DNA sequencing of the forward strand of each fragment was performed at the Eurofins Genomics(Germany). The resulting sequences were aligned in CLUSTALW46 using the program BioEdit47.

How were the areas covered by the different organisms measured?

Reply: The areas covered by fungi from the Mucorales group was measured using a transparent plastic format as shown In Appendix B. Each area has defined surface size. By placing this transparent plastic format under bottom the plates with fungal colonies, the area covered by non- A. fumigatus (fungi of Mucorales group) was estimated by adding up all grids covered by these colonies. This information has been inserted in the text in Line 154-158.

RESULTS

Abbreviations such as CFU and Cyp51 need to be specified the first time they appear in the text. (there are several other, like FM, MM, etc).

Reply: Thank you for your comments, this is has been adjusted in Line 58, Line195

Just as DG18 and M-RB media were incubated longer (5 days) than the other to achieve growth, it would be informative to culture MEA+C+S and SDA+C for shorter periods of time than 3 days to allow distinguishing between the colonies.

Reply: The growth rate of Mucorales is much higher than of A. fumigatus. Due to this rapid growing of Mucorales, even though after 2 days, some A. fumigatus colonies were covered by fast growing Mucor. Therefore, still we could quantitatively analyse the amount A. fumigatus in the samples in an unbiased way.

In Fig 2, there seem to be colonies different from both Aspergillus and Mucorales, by size or colour, and it would be interesting to know whether these are indeed different organisms or strains or not. A more detailed description of this section would be helpful.

Reply: Thank you for your comments. Indeed, Fig 2 showed the variation in size and colour in Aspergillus and Mucorales on different media, this is can be explained by the ingredients or selection of each medium, leading to various macroscopic phenotypes of these fungi. Actually all A fumigatus are the same mixed population(can be mixed genotypes or not depends on sample) on different media because all these come from a well- mixed compost sample. Fungi from the Mucorales group might appear differently on each medium, because different media may only inhibit certain species of Mucorales, but allow another types to grow, since (unlikely A. fumigatus) Mucorales represents a wide variety of species.

The last sentence of the first paragraph in section 3.3 doesn’t make sense, would these compounds be essential to stop Mucorales from growing? This needs to be explained more clearly. Similarly, the before last sentence of the next paragraph states that the addition of antibiotics makes 10% of the samples Mucorales-free. Does this refer to 10% of the previous 50% or in general, this needs to be explained better.

Reply: Thank you for your comments. We agree that in the previous version this section on what compounds are essential in achieving the inhibitive effects was not clear. This section has been re-written as follows:

Flamingo Medium can efficiently and selectively isolate A. fumigatus from environmental samples and this medium effectively inhibited growth of 95 % Mucorales from agricultural samples when these were plated on the Flamingo Medium, illustrating that the Flamingo Medium is selective. To investigate which element is critical for inhibiting the growth of Mucorales in Flamingo Medium, RB and Dichloran were removed step by step. As shown in Figure 3B and 3C, Mucorales were able to grow on the medium when either RB or Dichloran was omitted, suggesting that both RB and Dichloran are essential components of the Flamingo Medium to achieve the inhibitory effect.

While on Flamingo Medium the growth of Mucorales fungi is greatly reduced, on 50% of the environmental samples (mainly soil, grass, wood chips) growth of Mucorales is still faintly visible, such as grass/root sample 143 ( blue arrow Figure 3D). After introducing the antibiotics of Chloramphenicol and Streptomycin into Flamingo Medium, for 22% out of 50% of environmental samples, growth of Mucorales was completely inhibited when plated. Therefore, we conclude that the combination of RB, Dichloran, Chloramphenicol and Streptomycin effectively inhibits growth of Mucorales species from the environmental samples and makes the Flamingo Medium completely selective. 

Section 4.3 is 1 sentence that is very long and hard to follow. It should be broken down into shorter statements and the results explained better.

Reply, we agree with the Reviewer and have re-written this section as follows (L....):

We prepared mixtures of Mucolares fungi with and without an A. fumigatus strain. These mixtures were plated on both DG18 and on Flamingo Medium. After incubation for three days at 48 °C, results show that Mucorales fungi grow on DG18 and not on Flamingo Medium, while A. fumigatus grows on both media. Results are illustrated in Figure 4.

I did not see any supplementary figures available on the system, only tables. A supplementary Fig 2 is mentioned in the first sentence of the results section.

All supplementary figures and tables have been transferred to Appendix A,B,C and are placed after the main text in Line 320-338.

Reviewer 2 Report

In this study, the authors compared 6 different media to cultivate environnemental samples in order to selectively isolate Aspergillus (and avoid extensive colonies of Mucorales). The developped "Flamingo medium" was effective to reach this objective. 

Abstract lines 18-20: sentence to modify

Materials and methods:

  • in the International System of Units, gram is abbreviated in g (instead of gm)
  • precise the use of tween 20 or tween 80 (lines 109, 116)
  • line 111: how long was centrifugated liquid?
  • How was measured "surface area covered by Mucorales species" (line 121)?
  • line 130: Supplementary Table 2 absent

Results:

  • line 190: did you mean 3C instead of 4C?
  • Figure 3D: there is a difference between the text and the legend of this figure

Supplementary material: Figure S1 "Flow of preparation showing how to prepare the Flamingo Medium" is not available

Author Response

REVIEW 2

In this study, the authors compared 6 different media to cultivate environnemental samples in order to selectively isolate Aspergillus (and avoid extensive colonies of Mucorales). The developped "Flamingo medium" was effective to reach this objective. 

Abstract lines 18-20: sentence to modify

Reply: This sentence has been revised to “Six agar media were compared for effectiveness in isolating Aspergillus fumigatus from agricultural plant waste, woodchip waste, green waste, soil, grass and air samples collected in The Netherlands at 48°C incubation.”

Materials and methods:

  • in the International System of Units, gram is abbreviated in g (instead of gm) has been adjusted
  • precise the use of tween 20 or tween 80 (lines 109, 116) has been added “tween 80”
  • line 111: how long was centrifugated liquid? Has been added” for 1 min
  • How was measured "surface area covered by Mucorales species" (line 121)?

Reply: The areas covered by fungi from the Mucorales group was measured using a transparent plastic format as shown in Appendix B. Each area has defined surface size. By placing this transparent plastic format under bottom the plates with fungal colonies, the area covered by non- A. fumigatus (fungi of Mucorales group) was estimated by adding up all grids covered by these colonies. This information has been inserted in the text in Line154-158

  • line 130: Supplementary Table 2 absent

Reply: All supplementary figures and tables have been transferred to Appendix A,B,C and are placed after the main text in L320-338.

Results:

  • line 190: did you mean 3C instead of 4C?

Reply: Thank you for pointing out, it should be 3C. This has been adjusted.

  • Figure 3D: there is a difference between the text and the legend of this figure,

Reply: Yes, they are the same, this paragraph has been re-written as follows:

While on Flamingo Medium the growth of Mucorales fungi is greatly reduced, on 50% of the environmental samples (mainly soil, grass, wood chips) growth of Mucorales is still faintly visible, such as grass/root sample 143 ( blue arrow Figure 3D). After introducing the antibiotics of Chloramphenicol and Streptomycin into Flamingo Medium, for 22% out of 50% of environmental samples, growth of Mucorales was completely inhibited when plated. Therefore, we conclude that the combination of RB, Dichloran, Chloramphenicol and Streptomycin effectively inhibits growth of Mucorales species from the environmental samples and makes the Flamingo Medium completely selective. 

Supplementary material: Figure S1 "Flow of preparation showing how to prepare the Flamingo Medium" is not available

Reply:  All supplementary figures and tables have been transferred to Appendix A,B,C and are placed after the main text in L303-315.

Round 2

Reviewer 1 Report

The new version of the manuscript has improved all the observations made in the first revision. The authors provided satisfactory answers to the questions and improved the text.

No further comments are put forward.